# The Genetic Determinants of Extreme UV Radiation and Desiccation Tolerance in a Bacterium Recovered from the Stratosphere

**DOI:** 10.3390/microorganisms13040756

**Published:** 2025-03-27

**Authors:** Adam J. Ellington, Tyler J. Schult, Christopher R. Reisch, Brent C. Christner

**Affiliations:** 1Department of Microbiology and Cell Science, Institute of Food and Agricultural Science, University of Florida, Gainesville, FL 32611, USA; 2Meso Scale Diagnostics, LLC, Rockville, MD 20850, USA; 3Genomatica, San Diego, CA 92121, USA

**Keywords:** UV radiation, desiccation, bioaerosols, stratosphere, DNA repair, oxidative stress

## Abstract

Microbes that survive transport to and in the stratosphere endure extremes of low temperature, atmospheric pressure, and relative humidity, as well as high fluxes in ultraviolet radiation (UVR). The high atmosphere thus provides an ideal environment to explore the genetic and physiological determinants conveying high tolerance to desiccation and UVR. In this study, we examined *Curtobacterium aetherium* L6-1, an actinobacterium obtained from stratospheric aerosol sampling that displays high resistance to desiccation and UVR. We found that its phylogenetic relatives are resistant to desiccation, but only *C. aetherium* displayed a high tolerance to UVR. Comparative genome analysis and directed evolution experiments implicated genes encoding photolyase, DNA nucleases and helicases, and catalases as responsible for UVR resistance in *C. aetherium*. Differential gene expression analysis revealed the upregulation of DNA repair and stress response mechanisms when cells were exposed to UVR, while genes encoding sugar transporters, sugar metabolism enzymes, and antioxidants were induced upon desiccation. Based on changes in gene expression as a function of water content, *C. aetherium* can modulate its metabolism through transcriptional regulation at very low moisture levels (*X_w_* < 0.25 g H_2_O per gram dry weight). Uncovering the genetic underpinnings of desiccation and UVR resistance in *C. aetherium* provides new insights into how bacterial DNA repair and antioxidant mechanisms function to exhibit traits at the extreme ends of phenotypic distributions.

## 1. Introduction

The ability of microbes to survive aerosolization and long-range transport in the atmosphere has important implications for human health [1,2], agriculture [3,4,5], meteorology [6,7,8], and astrobiology [9,10]. Most studies of microbial aerosols have focused on those in the lowest layer of the atmosphere (i.e., troposphere) and where well-mixed air masses directly interact with the surface (i.e., convective boundary layer, CBL). Considering this, reports documenting microbial assemblage compositions in samples from the CBL that are distinct from those in overlying free tropospheric air masses [11,12] are unsurprising. In addition to differences in composition, emission sources, and the residency time of entrained aerosols, the environmental conditions in the CBL are not representative of those in higher portions of the atmosphere, where biological stresses (e.g., water loss and ultraviolet radiation, UVR) intensify with altitude. Though 10^23^ to 10^24^ microbial cells are estimated to disperse annually in the Earth–atmosphere system [13], there is a very limited understanding of the molecular and physiological mechanisms that enable survival under the conditions associated with high altitudes and long-range atmospheric transport.

The stratosphere (~15 to 50 km above sea level, ASL) is the atmospheric layer above the troposphere, and relative to conditions near the surface, it is characterized by low temperature, atmospheric pressure, and relative humidity (RH) and high UVR flux [9,12,13,14,15,16]. Consequently, any microbe surviving transport to and in the stratosphere must be highly tolerant of water loss and UVR [14]. Of the three types of UVR in the electromagnetic spectrum [UVAR (320–400 nm), UVBR (280–320 nm), and UVCR (100–280 nm)], UVCR is a significant portion of the solar radiation incident on Earth’s upper atmosphere. However, only UVAR and UVBR are transmitted to the surface because oxygen and ozone strongly absorb UVCR in the lower stratosphere [15]. As such, the stratosphere represents the only natural laboratory on Earth to examine the biological effects of highly mutagenic UVCR wavelengths. The combination of physical conditions in the stratosphere is unique compared to every other environment on Earth and highly similar to environmental settings at surface pressures on Mars [16,17]. For this reason, Earth’s stratosphere provides a tractable astrobiological analog that allows us to evaluate microbial survival potential in alien atmospheres [18,19,20,21].

UVR is biologically damaging through its direct and indirect effects on cellular macromolecules. The direct absorption of UVR by DNA forms pyrimidine dimers that inhibit transcription and replication, and can lead to single-stranded breaks in the sugar–phosphate backbone [22]. The indirect effects of UVR result from the photolysis of water molecules, which generates reactive oxygen species (ROS) that can induce single- and double- strand breaks, apurinic sites, and base damage in DNA, as well as the oxidation and chemical modification of proteins and lipids [23]. Research on bacteria over the last century has revealed that they have a complex genetic and physiological response to UVR exposure (termed the “UV-resistome”) that consists of five basic components: sense, shield, detoxify, repair, and tolerate [24]. Photoreceptors and other stress sensors sense specific UV wavelengths and DNA damage, triggering a regulatory cascade that elicits specific cellular responses [25,26]. Specialized membrane proteins and UV-absorbing pigments shield intracellular components from irradiation [27]. ROS-scavengers detoxify reactive species to prevent oxidative damage and maintain cellular redox homeostasis [28]. Efficient DNA repair proteins repair genetic damage [29]. And finally, error-prone polymerases tolerate and bypass unrepaired lesions to ensure cellular survival at the cost of genome fidelity [30]. The rich history of experimental work on model organisms such as *E. coli* has provided a basis for understanding the general response of bacteria to UVR and the DNA repair strategies utilized [22,29,31]. However, few efforts have focused their study on bacteria more representative of the species and environments that experience high exposures of UVR in nature.

When bacteria lose water during desiccation, they experience stresses that range from the physical disruption of cells to chemically inhibited enzymatic and metabolic activities. Water loss induces a reduction in cell volume, leading to the shrinkage of the cell membrane, overcrowding of macromolecules, decreased fluidity of the cytosol, denaturation of proteins, and an increase in the concentration of metabolites and ions [32,33]. Tolerance to water loss can be provided by accumulating ions (e.g., K^+^) and osmoprotectants (e.g., trehalose, sucrose, glycine betaine, etc.) that balance osmotic pressure and stabilize the protein structure. Desiccation also induces oxidative stress that can be mitigated by expressing ROS-scavenging enzymes, such as catalases and peroxidases [34,35,36], as well as non-enzymatic antioxidants that include pigments [37,38,39,40] and low-molecular-weight thiols [41,42,43]. Considering that resistance to oxidative stress could manifest tolerance to multiple extreme conditions, we sought to explore this possibility in a bacterium from an environment that selects for extreme UVR- and desiccation-resistant phenotypes.

Previously, we sampled microbial aerosols at altitudes up to 38 km ASL using customized helium balloon payloads [44], cultured and identified various bacteria, and characterized isolates with high tolerances to desiccation (15 to 25% RH) and UVCR (λ = 254 nm) [14]. The objective of this research was to conduct a genetic and physiological study of *Curtobacterium aetherium* L6-1 [45] in order to identify potentially novel protection and repair strategies possessed by bacteria that can survive conditions in the stratosphere. Though a landscape of genetic processes likely contributes to survival under individual stressors in the atmosphere, including highly efficient DNA repair mechanisms, we examined the possibility that extreme resistance to UVC radiation and desiccation were linked through mechanisms that mitigate oxidative stress. We established that *C. aetherium* possesses efficient cellular ROS detoxification mechanisms and used comparative genomic analysis and directed evolution to identify candidate genetic features contributing to the expressed characteristics. To assess our functional predictions, we examined the transcriptional response of *C. aetherium* to UVCR exposure and desiccation stress. Lastly, we used an in vivo photorepair assay to evaluate the physiological advantages and synergistic effects afforded by the activities of two classes of DNA repair proteins: cyclobutane pyrimidine dimer (CPD) and pyrimidine (6-4) pyrimidone photoproduct (6-4PP) photolyases. Together, our results lay the groundwork for understanding the genetic basis of *C. aetherium*’s extreme phenotypes. We discuss the contribution of our findings to bacterial UVR photobiology and deliberate on the pleiotropic effects afforded by mechanisms that enable microbial survival under prolonged water stress.

## 2. Materials and Methods

### 2.1. Bacterial Strains and Culture Conditions

The bacterial strains used in this study are listed in Appendix A. *C. aetherium* L6-1 [45] was isolated from aerosol samples collected using a helium balloon payload that sampled at altitudes of 18 to 23 km ASL near Ft. Sumner, New Mexico in 2013 [44]. To enable a comparative approach, reference strains phylogenetically related to *C. aetherium* and available in public culture collections [the German Collection of Microorganisms and Cell Cultures (DSMZ) and the United States Department of Agriculture (USDA) Agricultural Research Service Culture Collection (NRRL)] or from researchers were obtained. These included *Curtobacterium flaccumfaciens* pv. *flaccumfaciens* (DSM 20129), *Mycetocola reblochoni* (LMG 22367), and *Plantibacter flavus* (DSM 14012T). Unless otherwise noted, bacteria were cultured aerobically at 30 °C with vigorous shaking in LB (10 g L^−1^ tryptone, 5 g L^−1^ yeast extract, 10 g L^−1^ NaCl) media. When required, media were amended with antibiotics at the following concentrations: spectinomycin (50 µg mL^−1^) and carbenicillin (100 µg mL^−1^).

### 2.2. UVCR Survival Assays

Cultures were grown in 5 mL of liquid media to stationary phase, and six serial dilutions of this material were prepared in 10 mM MgSO_4_. Ten μL of each dilution was transferred to six sections of a square plate containing agar-solidified growth media. Portions of the plate were covered with aluminum foil such that sections uncovered sequentially during 1 min interval exposures were exposed to controlled doses of UVCR (GE G36T5 UVC Light Bulb; λ = 254 nm; 3.3 W m^−2^) that were 198, 396, 594, 792, and 990 J m^−2^. The UVCR doses were measured using a digital radiometer (Solar Light Company, Inc., Glenside, PA, USA). The longest exposure of the populations was 5 min, and the section of the plate that remained covered during the experiment (i.e., not exposed to UVCR) served as the control (N_0_). After exposure, the cultures were incubated at 30 °C for 24 to 48 h and the number of colony forming units (CFUs) surviving each UVCR dose (N) was used to calculate the surviving fraction, expressed as N/N_0_. Survival rates are reported as the mean and SEM for three biological replicates.

### 2.3. Whole-Genome Sequencing and Comparative Genomics

Whole-genome sequences for *C. aetherium* L6-1 and *C. flaccumfaciens* DSM 20129 were obtained, as previously described [46]. Briefly, stationary-phase cultures were pelleted, frozen (−70 °C), and shipped to SNPsaurus (Eugene, OR, USA) for DNA extraction, library preparation, sequencing, and genome assembly using the PacBio Sequel II sequencing platform followed by de novo genome assembly using the Flye v2.7 assembler [47]. The assembled genomes were submitted to GenBank (L6-1: CP076544.1; DSM 20129: CP080395.1/CP080396.1) and annotated using the NCBI Prokaryotic Genome Annotation Pipeline (PGAP v5.2) [48].

The average nucleotide identity (ANI) and average amino acid identity (AAI) among the isolate and reference strains were calculated using the ANI and AAI calculators from the Kostas lab (http://enve-omics.ce.gatech.edu/; accessed on 8 November 2022) [49,50]. Functional comparisons were made with the Rapid Annotation using Subsystem Technology (RAST) server and the SEED Viewer [51,52,53]. Further comparisons were made by searching for homologs of known DNA repair and ROS detoxification proteins in the L6-1 and DSM 20129 genomes using the blastp algorithm from NCBI’s Basic Local Alignment Search Tool (BLAST v2.13.0) [54]. Query protein sequences were obtained from the UniProt database for *Escherichia coli* strain K12, *Bacillus subtilis* strain 168, and *Mycobacterium tuberculosis* strain ATCC 25618/H37Rv [55]. Reciprocal best hits were considered homologs if the bit score was >50, E-value was <1 × 10^−5^, and query coverage was >50% [56]. Differences in gene content among the isolates and the reference strains were visualized using the BLAST Ring Image Generator (BRIG) [57].

### 2.4. ROS Quantification Assays

The free radical probe 2’,7’-dichlorodihydrofluoresceindiacetate (H_2_DCFDA) (Biotium, Fremont, CA, USA) was used to quantify ROS. Stationary-phase cultures of *E. coli* MG1655, *D. radiodurans* R1, isolate L6-1, the wild-type parent strain DSM 20129, and UV-evolved strain DSM-9.3.3 were diluted to OD_600_ 1.0 in 10 mL of media. The cells were washed with 10 mM potassium phosphate buffer (pH 7.0) and incubated for 30 min at 25 °C in the same buffer containing 10 μM H_2_DCFDA. Subsequently, cells were washed and resuspended in 10 mL potassium phosphate buffer (10 mM; pH 7.0), transferred to 60 × 15 mm Petri plates, and exposed to UVCR doses of 0, 990, or 1980 J m^−2^ while being stirred at 400 rpm with a magnetic stir bar. The exposed cell populations were then washed, resuspended in fresh buffer, and disrupted by vortexing with lysing matrix B (MP Biomedicals, Santa Ana, CA, USA). Cellular debris was removed by centrifugation for 10 min at 5000× *g*, and fluorescence intensity in the supernatant was measured using a multi-well plate reader (SpectraMax M3; Exc. 490 nm, Emm. 519 nm; Molecular Devices, San Jose, CA, USA). Fluorescence intensity was normalized per mg of protein, as determined by the Pierce BCA Protein Assay Kit (ThermoFisher Scientific, Waltham, MA, USA), and compared relative to data from unexposed controls (i.e., time zero is 100% fluorescence intensity). The values reported are the averages of three biological replicates and the error bars indicate SEM.

### 2.5. Photolyase Activity Assay

To assess in vivo photorepair, isolate L6-1, the DSM 20129 founder strain, and the UV-evolved strain DSM-9.3.3 were grown to stationary phase and exposed to UVCR using the same method as described for the UVCR survival assays. After exposure, half of the cultures were immediately placed in the dark and the other half were photoreactivated for 1 h under a daylight lamp (GE F20T12-C50-ECO 20W Chroma 50 5000K Ecolux Daylight Light Bulb; General Electric Company, Cincinnati, OH, USA). All the cultures were subsequently incubated in the dark at 30 °C until colonies formed. Survival rates under dark and light conditions were calculated, and the lethal dose reducing the population by 90% (LD_90_) was determined by fitting the survival curves to an exponential decay model. The percentage increase in LD_90_ by photoreactivation was calculated using the following equation: $\frac{{LD90}_{Light}-{LD90}_{Dark}}{{LD90}_{Dark}}\times 100=\%\;Increase$. Results are reported as the averages of four biological replicates, with error bars representing SEM.

### 2.6. UVCR and Desiccation Exposure for RNAseq Experiments

Triplicate cultures of *C. aetherium* L6-1 were grown aerobically (250 rpm) at 30 °C to their stationary phase in 50 mL of LB media (10 g L^−1^ tryptone, 5 g L^−1^ yeast extract, 10 g L^−1^ NaCl) in a 250 mL flask. The cultures were diluted to OD_600_ 0.5 in 10 mL of fresh media.

For UVCR exposure, the diluted cultures were transferred to 60 × 15 mm Petri plates and irradiated with 1980 J m^−2^ of UVC light (GE G36T5 UVC Light Bulb; λ = 254 nm; 3.3 W m^−2^; General Electric Company, Cincinnati, OH, USA) while being stirred at 400 rpm with a magnetic stir bar. Samples (500 μL) were taken before exposure (t = 0), immediately after 10 min of UVCR exposure, and after a 20 min recovery period. The media were removed after centrifugation at 5000× *g* for 2 min, and the cell pellets were resuspended in 1X DNA/RNA Protect Reagent (New England Biolabs; Ipswich, MA, USA). The cell suspensions were then immediately lysed by bead-beating with lysing matrix B (MP Biomedicals, Santa Ana, CA, USA) in a laboratory mixer mill (Retsch Mixer Mill MM 400, Verder Scientific Inc., Newtown, PA, USA) at 30 Hz for 5 min. Cellular debris was removed by centrifugation for 2 min at 16,000× *g* and supernatants were stored on ice until RNA extraction was performed.

For experiments that examined the response to desiccation, 500 μL of diluted stationary-phase cultures (OD_600_ 0.5) were spotted onto four 0.2 μm pore size filters that were placed into 60 × 15 mm Petri plates. The initial masses of the samples were determined using an analytical balance (Sartorius Practum 224-1S; Göttingen, Germany), and the samples were placed in a desiccation chamber (SP Bel-Art Secador 4.0 Auto-Desiccator, BelArt, Wayne, NJ, USA) that maintained RH levels between 25 and 30%. At designated time points, each sample was weighed to calculate water loss. Samples were removed from the desiccation chamber upon reaching approximately 75%, 50%, 25%, and 0% of their initial water content. Cells were resuspended from the filters by vortexing in 1X DNA/RNA Protect Reagent (NEB) and immediately lysed by bead-beating as described above. Cellular debris was removed by centrifugation for 2 min at 16,000× *g* and the supernatants were stored on ice until RNA extraction was performed.

### 2.7. RNA Extraction and Sequencing

Total RNA was isolated from the UVCR- and desiccation-exposed samples using the Monarch Total RNA Miniprep Kit (New England Biolabs; Ipswich, MA, USA) and following the manufacturer’s recommended protocol. Genomic DNA was removed using a gDNA removal column in addition to treatment with DNase I when bound directly to the column matrix. Purified RNA was quantified and RNA integrity was determined on a Qubit 4 Fluorometer using the Qubit RNA High Sensitivity Assay Kit (ThermoFisher Scientific, Waltham, MA, USA) and the Qubit RNA IQ Assay Kit (ThermoFisher Scientific), respectively. Purified RNA was stored at −70 °C until preparation for sequencing.

The presence and quantity of RNA molecules were determined by sequencing (i.e., RNA-seq). The raw RNA samples were thawed on ice, followed by rRNA depletion, cDNA synthesis, adaptor ligation, and library amplification using the Illumina Stranded Total RNA Prep Kit with Ribo-Zero Plus (Illumina, San Diego, CA, USA) and following the manufacturer’s recommended protocol. Prepared sequencing libraries were pooled and sequenced on the Illumina NovaSeq 6000 sequencing platform with 2 × 150 bp paired-end reads (Novogene Co., Sacramento, CA, USA).

### 2.8. Preprocessing, Mapping Sequencing Data, and Differential Expression Analysis

The reads obtained from RNA-seq were quality filtered and the adaptors were trimmed using the Trim Galore script v0.6.7 [58], followed by mapping to the *C. aetherium* L6-1 genome (NCBI Reference Sequence: NZ_CP076544.1) with Bowtie2 v2.4.5 using the default parameters [59]. Reads mapping to annotated genomic features were computed using the featureCounts function in the Rsubread package (v1.22.2), with the -B and --countReadPairs flags set to only consider concordantly mapped read pairs [60]. Read pairs mapping to rRNA regions were not included in further analyses.

Differential gene expression analysis was conducted using the DESeq2 program [61]. DESeq2 uses a negative binomial generalized linear model and an internal normalization that accounts for sequencing depth. Samples were compared pairwise by looking at the expression levels under the UVCR and desiccation conditions versus the control samples (t = 0). Genes with fewer than 10 total raw counts were removed prior to running DESeq2, and additional independent filtering parameters were kept at DESeq2’s defaults. The DESeq2 package uses a Wald test to determine the statistical significance of the differential expression results, and the Benjamini–Hochberg correction is used to correct for multiple testing, yielding an adjusted *p*-value. The shrinkage estimation of log_2_ fold changes was used to control for technical and biological variability without the need for arbitrary filters on low-count genes [61], and the apeglm shrinkage estimator [62], available within the DESeq2 package, was applied to our data. Gene expression data with an adjusted *p*-value < 0.05 and log_2_ fold change ≥ 2 or ≤−2 were considered significantly different. A complete list of the log_2_-fold changes (log_2_FC) and adjusted *p*-values (*p*.adjust) for all the genes in the *C. aetherium* L6-1 genome under each condition is provided in Appendix A. Using these data, volcano plots were created using the EnhancedVolcano [63] and ggplot2 v 3.4.0 [64] R packages.

### 2.9. Gene Set Enrichment Analysis of Differentially Expressed Genes

Gene Ontology (GO) functional annotations for all the protein coding genes in the *C. aetherium* L6-1 genome were assigned using the Blast2GO v6.0 program [65], with the GO-slims option in order to reduce redundancy of terms. A custom OrgDB package was then created for strain L6-1 using the AnnotationForge R package [66], and gene set enrichment analysis (GSEA) was performed with clusterProfiler using the gseGO function [67,68].

## 3. Results

### 3.1. UVCR Resistance in C. aetherium and Phylogenetically Related Bacteria

Based on the comparison of the 16S rRNA gene sequence from strain L6-1, this isolate was initially identified as a member of the Gram-positive actinomycetotal genus *Curtobacterium* [14]. To assess whether the high UVCR tolerance L6-1 displays was a feature shared with other closely related taxa, its 16S rRNA gene sequence was used to search public databases for closely related Actinomycetota. This analysis identified several close relatives of L6-1 (Figure 1A; Appendix A) that are maintained in public culture collections (e.g., ATCC, DSMZ, USDA), which were subsequently obtained and screened for UVCR tolerance. The inactivation rate (*k*) and LD_90_ were derived from a survival curve for each strain (Appendix A). L6-1 had nearly five-fold higher UVCR tolerance (LD_90_ of 470 J m^−2^) than DSM 20129 (LD_90_ of 98 J m^−2^; Figure 1B; Appendix A).

### 3.2. Genetic Differences Between UVCR-Tolerant and -Sensitive Strains

The disparities in UVCR tolerance between the stratospheric isolate and reference strains suggested that L6-1 possesses genetically discernable features contributing to this phenotype. To enable comparative genomic analyses, complete whole-genome sequences were obtained for the *Curtobacterium* strains and deposited in GenBank [46].

L6-1 (i.e., *C. aetherium*; [45]) has a single 3.4 Mbp circular chromosome with a GC content of 72.0% and 3158 genes encoding 12 rRNAs, 46 tRNAs, and 3072 proteins (Table 1). *C. flaccumfaciens* DSM 20129 possesses a circular chromosome and plasmid that together total 3.8 Mbp, with a GC content of 70.9%, and 3616 genes encoding 9 rRNAs, 47 tRNAs, and 3517 proteins. The two strains share an 84% average nucleotide identity (ANI), which is well below the >95% values typically observed for closely related populations that are considered members of the same species taxon [69,70,71,72].

To analyze the DNA repair and ROS detoxification pathways of the UV-resistome, genome comparisons using the RAST annotation server and the BLAST alignment tool were performed (Appendix A). *C. aetherium* L6-1 encodes four additional proteins associated with DNA repair that are not found in DSM 20129 (Figure 2; Appendix A), including endonuclease VIII (*nei2*), two homologs of the UvrD/PcrA DNA helicases (*uvrD2* and *pcrA*), and a cryptochrome/photolyase family protein (*phr1*). In addition to DNA repair genes, L6-1 encodes a homolog of the KatA catalase (*katA*; Figure 2; Appendix A), the major catalase expressed in vegetative *Bacillus* [35], whereas DSM 20129 encodes a KatX homolog (Appendix A) that has been demonstrated to protect germinating endospores from H_2_O_2_ stress [73]. A homolog of the peroxide operon regulator PerR (*perR*) from *B. subtilis* is also found in L6-1, but not DSM 20129 (Figure 2; Appendix A). L6-1 also possesses an extra copy of the glutaredoxin-like protein NrdH (*nrdH2*; Figure 2; Appendix A) involved in disulfide reduction.

### 3.3. Directed Evolution of UVCR Tolerance in C. flaccumfaciens DSM 20129

The experiments designed to improve the UVCR tolerance of the reference strain through repeated exposure to sublethal doses of UVCR (Appendix A) were partially successful (Appendix A). Decreased UVCR sensitivity occurred incrementally in strain DSM 20129 during 12 rounds of selection, increasing its LD_90_ 3.5-fold in comparison to the founder population and to a level comparable with isolate L6-1 (Appendix A). Whole-genome sequences were obtained for five evolved strains of DSM 20129, selected at points in the experiment where increased UVCR tolerance evolved in the populations (Appendix A). The mutations acquired were identified using the Breseq mutation analysis pipeline [74]. In total, 40 mutations were identified in the most tolerant strain (round 9; DSM-9.3.3), with 5 occurring in intergenic regions, 6 in hypothetical genes, 3 in pseudogenes, and 26 in protein coding genes (Appendix A). Of the 26 mutations occurring in protein coding genes, 6 are silent and 20 are nonsynonymous mutations (Appendix A) for various protein-coding genes (Appendix A). The most notable are involved in DNA repair (uracil DNA glycosylase and DNA photolyase), cellular redox homeostasis (NAD(P)/FAD-dependent oxidoreductase in the thioredoxin reductase family), and the stress response (cold-shock protein). The most tolerant evolved strain (DSM-9.3.3) was selected for use in subsequent experiments.

### 3.4. Intracellular ROS Concentrations After UVCR Exposure

The indirect effects of UVR exposure are well established, and our analysis implicated various genes involved in oxidative stress that may contribute to the high levels of UVCR tolerance observed. We examined cellular ROS concentrations to determine if the capacity to efficiently detoxify ROS correlated to UVCR tolerance. ROS concentrations before and after exposure to UVCR were monitored for L6-1, the DSM 20129 parent strain, and the most UVCR-tolerant evolved strain (DSM-9.3.3) using the free radical-sensing fluorescent probe H_2_DCFDA. ROS concentrations did not significantly increase in L6-1, DSM 20129, or DSM-9.3.3 at any of the UVCR doses tested (up to 1980 J m^−2^; Figure 3). Identical observations were made in experiments with *Deinococcus radiodurans* R1 (Figure 3), which is well known for its capacity to resist and detoxify oxidative stress-generating agents [75,76,77]. In contrast, ROS concentrations in *E. coli*, which is not highly tolerant of oxidative stress, approximately doubled after each ~1000 J m^−2^ of UVCR exposure (Figure 3).

### 3.5. In Vivo Assessment of Photolyase Activity

The contribution of photoreactivation to UVCR tolerance was investigated using an in vivo photorepair assay. Light activates photolyase for the repair of DNA photoproducts [78]; therefore, its relative activity can be determined by examining the survival rate of UVCR-exposed populations that recover under white light in comparison to those kept in the dark. Consistent with previous studies [79,80,81], post-UVCR exposure to white light increased the survival rate for all strains (Figure 4A). Based on interpolated LD_90_ values, there was not a significant difference in survival for DSM 20129 and DSM-9.3.3, indicating that the mutations incurred in the photolyase gene did not significantly affect DNA repair activity under the conditions tested (Figure 4A inset). However, the photoreactivation of L6-1 increased the LD_90_ by ~140%, which is significantly higher (*p* = 0.004; one-way ANOVA) than the ~19% increase observed for the DSM strains (Figure 4A inset).

To explore if the two photolyase genes found in *C. aetherium* L6-1 could contribute to its superior photorepair capacity, we aligned the sequences to known CPD and (6-4) photolyases. The CPD photolyases from *E. coli* (EcPhrB) and *Agrobacterium fabrum* (AfPhrA) align well with one of the photolyases found in L6-1 (*phr2*: KM842_RS02105) and the photolyase from DSM 20129 (*phr*: K0028_16065) (Appendix A). However, the additional photolyase gene in L6-1 (*phr1*: KM842_RS01465) is more closely related (~37% identity) to the bacterial (6-4) photolyases of *A. fabrum* (AfPhrB), *Cereibacter sphaeroides* (CsCryB), and *Vibrio cholerae* (Vc(6–4) FeS-BCP); the active site residues determined for AfPhrB are largely conserved in all four species (Appendix A). The protein structural prediction algorithm AlphaFold2 was used to model 3D structures for the L6-1 and DSM 20129 photolyases with a high degree of confidence (Appendix A). The 3D structural alignments with the *E. coli* CPD photolyase (EcPhrB, PDB ID: 1DNP) show high similarity with the L6-1 Phr2 and DSM 20129 Phr, with a template-modeling score (TM-score) of 0.87 for both proteins (Appendix A). The L6-1 Phr1 shows high similarity with the *A. fabrum* (6-4) photolyase (AfPhrB, PDB ID: 5KCM), with a TM-score of 0.89, further supporting its proposed function as a bacterial (6-4) photolyase (Appendix A).

To assess the contributions of *phr1* and *phr2*, expressed individually and in combination, to survival under UVCR exposure, we cloned them into expression vectors for heterologous expression in *E. coli* BL21 cells. Wild-type BL21 cells were rapidly inactivated upon UVCR exposure, with an LD_90_ of 16 J m^−2^ under photoreactivating conditions (Appendix A). The overexpression of the native *E. coli* CPD photolyase (EcPhrB) significantly increased the LD_90_ by more than 230% during photoreactivation (Figure 4B). Photoreactivation by the putative CPD photolyases from the *Curtobacterium* strains (L6Phr2 and DSMPhr for L6-1 *phr2* and DSM 20129 *phr*, respectively) also improved the survival of BL21, increasing the LD_90_ by 330% and 270%, respectively (Figure 4B). In contrast, expression of the putative (6-4) photolyase from strain L6-1 (L6Phr1) alone did not significantly improve the UVCR survival of BL21 cells (Figure 4B). However, when the two L6-1 photolyases were co-expressed (L6Phr1 + L6Phr2), photoreactivation significantly improved survival over L6Phr2 alone (Figure 4B) and increased the LD_90_ by an average of 400%.

### 3.6. Differential Expression Analysis

To examine changes in gene expression and determine whether the additional genes possessed by *C. aetherium* L6-1 are expressed in response to UVR and desiccation, whole-transcriptome sequencing was performed. Data from three independent replicate cultures that were UVC irradiated (with and without a 20 min recovery period) or desiccated (to approximately 75%, 50%, 25%, and 0% of their initial water content) were compared to the control samples. On average, approximately 16 million raw reads were obtained for each sample (Appendix A). After adaptor trimming and quality filtering, 88 to 96% of reads mapped against the *C. aetherium* L6-1 reference genome, of which, an average of 94% were assigned to genomic features among samples. The remaining 6% of the reads were discarded because they aligned to an intergenic region or overlapped with two or more genomic features. The number of curated reads mapped and assigned to genomic features ranged from 12 to 14.5 million per sample (Appendix A).

The Ribo-Zero Plus method was much less efficient in depleting the rRNA transcripts for strain L6-1 in comparison to the model species (i.e., *E. coli* and *Bacillus subtilis*) tested by [82], and, despite enzymatic rRNA depletion, rRNA transcripts represented 62 to 76% of the total reads. After the removal of the rRNA sequences, an average of 2.7 to 4.4 million reads per sample were retained for further analysis and classed as either a coding sequence (CDS), tRNA, tmRNA, or ncRNA (Appendix A). Reduced sequencing depth can be compensated for by an experimental design that includes two or more biological replicates [83,84], and, thus, the observations we obtained from independent triplicate samples provided robust data for statistical analysis.

Comparative transcriptome analysis was performed using a cutoff value of |log_2_FC| > 2 (i.e., gene expression was 4 times higher or lower in one group compared with another) and a *p*.adjust < 0.05 to determine if the differentially expressed gene (DEG) data were statistically significant. Examining the gene expression data using principal component analysis (PCA) showed that 84% of the variance is explained by principal components 1 and 2, and that replicate samples cluster by condition (Appendix A). This indicates that the expression profiles observed are better explained by the stress variable than by variation among replicates.

### 3.7. Transcriptional Changes in Response to UVCR Exposure

For the UCVR-treated samples of *C. aetherium* L6-1, 82 total DEGs were identified (81 upregulated and 1 downregulated) immediately following UVCR exposure (‘UVC-only’; Figure 5A). After a 20 min period of recovery (‘UVC + rec’), there were a total of 141 DEGs, with 124 upregulated and 17 downregulated. Only 10 DEGs were unique to the UVC-only samples, while 69 were unique to the UVC + rec samples; 72 DEGs were shared between both conditions (Figure 5A).

The statistical threshold applied to the DEG analysis enabled identification of the most highly altered gene expression levels in the experiments. However, phenotypic differences may be manifested by smaller, consistent changes in the expression of a group of genes. Such coordinated patterns of expression in our experiments were examined by GSEA, which aggregates per-gene statistics across a set of genes [85]. To assess whether the expression of specific categories of genes was altered by UVCR exposure, we assigned functional categories to each gene in the L6-1 genome based on Gene Ontology (GO) terms and performed GSEA for each condition. For both the UVC-only and UVC + rec samples, the top enriched gene sets activated upon UVCR exposure were related to DNA metabolic processes, DNA repair, and stress response (Figure 5B). Notably, approximately 60% of the genes related to DNA metabolism and/or repair and about 50% of those related to stress response were activated under both conditions (Figure 5B). In the UVC-only samples, the top gene sets suppressed under UVCR exposure were carbohydrate metabolisms, nucleobase-containing small molecule pathways, signaling, and the generation of precursor metabolites and energy. The UVC + rec samples also showed suppression of organic acid metabolic processes (e.g., amino acids, carboxylic acids, etc.) (Figure 5B), consistent with previous reports of hindered metabolic activity following exposure to UVR [21,86].

An exhaustive list of the DEGs for the UVCR-treated samples is provided in Appendix A, and the sections that follow summarize the most statistically significant DEGs in the top activated gene sets: DNA metabolism/repair, stress response, and regulation of transcription/translation.

#### 3.7.1. DNA Metabolism and Repair in Response to UVCR

In the UVC-only condition, a total of 24 genes with known or potential roles in DNA metabolism and repair were differentially expressed (Figure 5C). Among them were two related to base excision repair (BER; Fpg/Nei family DNA glycosylase and DNA-3-methyladenine glycosylase 2), two related to nucleotide excision repair (NER; UvrA and UvrB), seven for recombination (RecA, RecX, RmuC, two RecQs, RadA, PD-(D/E)XK nuclease family protein, and GIY-YIG nuclease family protein), one for translesion synthesis (DNA polymerase IV), and the DNA ligase LigA. There was also upregulation of the replication proteins DNA polymerase I and DNA polymerase III subunits gamma and tau, type II DNA topoisomerase, and two helicases (replicative DNA helicase and ATP-dependent helicase), as well as three exonucleases (SbcD, a 3’-5’ exonuclease, and an exonuclease domain-containing protein), transcription repair coupling factor, and a spore photoproduct lyase family protein (Figure 5C).

All the genes in the UVC-only treatment were also upregulated in the UVC + rec condition, many of which exhibited higher log_2_FC values for expression (Figure 5D), except for DNA polymerase I, which fell just short of the cutoff (log_2_FC = 1.90). The UVC + rec samples also showed an upregulation of ten additional genes related to DNA metabolism and repair: two error-prone polymerases (DNA polymerase Y family protein and error-prone DNA polymerase), UvrC, RecR, RecG, DNA polymerase III subunit delta, a bifunctional 3’–5’ exonuclease/DNA polymerase, a methylated-DNA-[protein]-cysteine S-methyltransferase involved in repairing alkylated guanine bases, and two additional types of Fpg/Nei DNA glycosylase and ATP-dependent DNA helicase (Figure 5D). Notably, photolyases were not included among activated DEGs. A closer examination of the two photolyases found in L6-1 showed that its CPD photolyase (KM842_RS02105) had a log_2_FC of 1.32 and 1.93 for the UVC-only and UVC + rec conditions, respectively, whereas its putative (6-4) photolyase (KM842_RS01465) showed log_2_FCs of 0.69 and 1.99, respectively (Appendix A). While the changes in expression data fall below threshold values for statistical significance, both photolyases were clearly upregulated in response to UVCR exposure.

#### 3.7.2. Stress Response and Regulation of Transcription and Translation

The UVC-only samples showed an upregulation of four genes potentially involved in the bacterial stress response and cell signaling: three peptidases (S26 family signal peptidase, S24/S26 family peptidase, and SOS response-associated peptidase) and a cold-shock binding domain protein (CsbD family protein) (Figure 5C).

Of the three peptidases upregulated under the UVC-only condition, only SOS response-associated peptidase remained upregulated in the UVC + rec samples (Figure 5D). In addition, the upregulation of an A24 family peptidase was observed under the UVC + rec condition (Figure 5D). Aspartic proteases of the A24 family have been shown to regulate bacterial competence [87], and the induction of competence regulons appears to occur as a general stress response in certain Gram-positive bacteria [88]. Indeed, we found the upregulation of an additional gene related to bacterial competence (ComEA family DNA-binding protein) in the UVC + rec samples.

The activation of three transcriptional regulators (transcriptional repressor, helix-turn-helix transcriptional regulator, and LysR substrate-binding domain-containing protein) was common to both UV conditions, with one specific to the UVC-only samples (MarR family transcriptional regulator) and one specific to the UVC + rec samples (metal-sensitive transcriptional regulator). In addition, two genes encoding GNAT family N-acetyltransferases were upregulated in both UV conditions, which are enzymes involved in the acetylation of proteins, a major regulatory post-translational modification in eukaryotes and prokaryotes [89].

### 3.8. Transcriptional Changes in Response to Desiccation Stress

The transcriptional response of *C. aetherium* L6-1 at varying degrees of water loss was examined to ascertain the molecular mechanisms it uses to prepare for desiccation. Samples were taken upon reaching 75%, 50%, 25%, and 0% of the initial water content (i.e., X_w_ = 0.75, 0.50, 0.25, and 0 g H_2_O per g of dry solid, respectively). In these experiments, “0%” water content represents the moisture level at which the cells reached equilibrium with the RH of the desiccation chamber (25–30% RH). When the samples were placed into the desiccator, it took approximately two hours to reach each of the designated sampling points, and the 0% H_2_O samples were allowed to equilibrate in the desiccation chamber for 24 h.

Differential expression analysis revealed that the number of DEGs increased with decreasing water availability. At 75% of the initial water content (Des_75), 68 DEGs were upregulated and 22 were downregulated (Figure 6A). This increased to 420 DEGs at 50% water content (Des_50), with 351 being upregulated and 69 downregulated. At 25% water content (Des_25), there were 512 DEGs (393 upregulated and 119 downregulated), and at 0% water content (Des_0), there were 595 DEGs (446 upregulated and 149 downregulated). There were 68 DEGs common to all levels of hydration, while 330, 127, and 119 were unique to the 50%, 25%, and 0% water content samples, respectively (Figure 6A).

GSEA showed that no gene sets were significantly enriched for the Des_75 samples. However, the samples at lower levels of water content showed the activation of gene sets related to carbohydrate metabolic processes, and ~30% of the genes in this category were upregulated (Figure 6B). Approximately 40% of genes related to the generation of precursor metabolites and energy were activated in the Des_50 samples. Many gene sets were highly suppressed at or below 50% water content, including ~80% of genes related to tRNA and ncRNA metabolic processes and cellular component biogenesis (i.e., biosynthesis of macromolecules), 50 to 60% of genes related to DNA replication, and 30 to 50% of genes related to metabolic processes for molecules other than carbohydrates (Figure 6B).

The following sections summarize the DEGs with known or putative functions in DNA metabolism and repair; stress response and the regulation of transcription and translation; nutrient transport and metabolism; and antioxidants and detoxification systems for oxidative stress. A complete list of the DEGs is provided in Appendix A.

#### 3.8.1. DNA Metabolism and Repair in Response to Desiccation

At all levels of hydration tested, two DEGs related to DNA metabolism and/or repair were upregulated: a RecQ family ATP-dependent DNA helicase and an endonuclease/exonuclease/phosphatase family protein (Figure 6C–F). The expression of these genes increased with decreasing water content, reaching log_2_FC = 5.9 and 7.6, respectively, in the Des_0 samples (Figure 6F). Proteins of the endonuclease/exonuclease/phosphatase family have diverse roles, including DNA repair for enzymes such as AP endonuclease [90]. At or below 50% water content, eight additional DEGs in this category were upregulated, including genes for a DNA starvation/stationary-phase protection protein, two proteins related to bacterial non-homologous end joining (NHEJ; a non-homologous end-joining DNA ligase and the Ku protein), two non-LigA ATP-dependent DNA ligases, an Fpg/Nei family DNA glycosylase involved in BER, a recombinase family protein, a cryptochrome/photolyase family protein, and a single-stranded DNA-binding protein (Figure 6D–F).

#### 3.8.2. Regulation of Stress Response, Transcription, and Translation

Two DEGs with roles in the stress response were upregulated at all four time points, including genes for an Asp23/Gls24 family envelope stress response protein and a GlsB/YeaQ/YmgE family stress response membrane protein. The samples at or below 50% water content also showed an upregulation of an additional six genes encoding for two CsbD family proteins, another Asp23/Gls24 family envelope stress response protein, another GlsB/YeaQ/YmgE family stress response membrane protein, a PrsW family glutamic-type intramembrane protease, and a trypsin-like peptidase domain-containing protein. Unique to the Des_0 samples were four DEGs for another CsbD family protein, a neutral zinc metallopeptidase, signal peptidase I, and an ice-binding family protein. Various transcriptional and translational regulators (six DEGs common to all samples) were activated under desiccation. This included genes for two unspecified RNA polymerase sigma factors, two response regulator transcription factors, and two MarR family transcriptional regulators. At or below 50% water content, an additional five DEGs were activated, that encode for a LuxR family transcriptional regulator, a helix-turn-helix transcriptional regulator, two N-acetyltransferases, and the ribosome-associated translation inhibitor RaiA.

At ≤25% water content, four more DEGs were upregulated, that encode a TetR/AcrR family transcriptional regulator, another LuxR family transcriptional regulator, another response regulator transcription factor, and the RNA polymerase-binding protein RbpA (Figure 6E,F). In the Des_0 samples, three more transcriptional regulators were upregulated: a third MarR family transcriptional regulator, a fourth response regulator transcription factor, and a DeoR/GlpR family DNA-binding transcriptional regulator (Figure 6F).

#### 3.8.3. Nutrient Transport and Metabolism

Six DEGs related to nutrient transport were upregulated at all levels of hydration tested: two ion transporters (a sodium/calcium antiporter and a cation-transporting ATPase) and four sugar transporters (a major facilitator family [MFS] transporter, two sugar ABC transporter permeases, and an ABC transporter substrate-binding protein). An additional two ion transporters (a ChaB family protein and mechanosensitive ion channel family protein), fifteen sugar transporters (nine carbohydrate/sugar transporter permeases, four ABC transporter substrate-binding proteins, an MFS transporter, and a PTS sugar transporter subunit IIA), an APC family permease, and a BCCT family transporter potentially involved in betaine uptake were upregulated at ≤50% water content. At or below 25% water content, the upregulation of another six transporters was observed: a Na^+^/H^+^ antiporter, two carbohydrate/sugar ABC transporter permeases, an inorganic phosphate transporter family protein, an ECF transporter S component, and the biopolymer transporter Tol. In the Des_0 samples, there were six more transporters activated: four ABC transporter permeases, an Nramp family divalent metal transporter, and another Na^+^/H^+^ antiporter. Notably, the gene encoding a manganese efflux pump MntP family protein was significantly downregulated in the Des_25 and Des_0 samples.

As anticipated, the expression of genes involved in metabolism was altered under desiccation stress. Genes for glycogen metabolism (glycogen debranching protein GlgX), the tricarboxylic acid (TCA) cycle (succinate–CoA ligase subunit alpha, ADP-forming succinate–CoA ligase subunit beta, citrate synthase), unspecified sugar metabolism (two glycosyltransferases, two glycosyl hydrolases, and a sugar phosphate isomerase/epimerase), and the cAMP-generating adenylyl cyclase were upregulated in all the samples. The downregulation of genes related to fatty acid biosynthesis (ACP S-malonyltransferase and ketoacyl-ACP synthase III), the thiol reductant exporter CydDC, and a cytochrome *d* involved in aerobic respiration were observed under all conditions.

At ≤50% water content, the further upregulation of genes for trehalose biosynthesis (malto-oligosyltrehalose synthase, malto-oligosyltrehalose trehalohydrolase, and maltose alpha-D-glucosyltransferase), fatty acid catabolism (acyl-CoA dehydrogenase, two long-chain fatty acid–CoA ligases, and a thiolase family protein), glycogen metabolism (1,4-alpha-glucan branching protein GlgB and another glycogen debranching protein GlgX), glucose generation from complex sugars or gluconeogenesis (alpha,alpha-phosphotrehalase, D-galactonate dehydratase family protein, alpha-amylase family protein, isocitrate lyase/PEP mutase family protein, and L-glyceraldehyde 3-phosphate reductase), the pentose phosphate pathway (phosphoketolase family protein, two gluconolactonases, and gluconokinase), glycolysis (PfkB family carbohydrate kinase and phosphoglycerate mutase family protein), and the TCA cycle (acetate–CoA ligase) was observed. Genes involved in flagellar assembly and motility (flagellar FlbD family protein, flagellin, and flagellar export chaperone FliS) were also downregulated in these samples.

Six additional genes were upregulated at a water content of ≤25% that encode proteins involved in the metabolism of various sugars (glycosyltransferase, 6-phospho-3-hexuloisomerase, alpha-xylosidase, mannose-1-phosphate guanylyltransferase, PQQ-dependent sugar dehydrogenase, and glycerophosphodiester phosphodiesterase) as well as one for fatty acid catabolism (3-hydroxyacyl-CoA dehydrogenase NAD-binding domain-containing protein). Concomitantly, 19 different genes encoding ribosomal proteins were downregulated in the samples, implying a ramping down of translation to prevent superfluous energy use. This contention is further supported by the downregulation of 12 different tRNAs in the Des_0 samples.

#### 3.8.4. Antioxidants and Detoxification Systems

Three DEGs encoding antioxidant proteins were upregulated under all conditions: an aldehyde dehydrogenase, a glutathione peroxidase, and an organic hydroperoxide resistance protein (Figure 6C–F).

At ≤50% water content, an additional 12 antioxidant-related DEGs were upregulated (Figure 6D–F). These included genes for nine aldo/keto reductases, a glutathione-dependent formaldehyde dehydrogenase involved in formaldehyde detoxification, and a manganese catalase family protein. Like aldehyde dehydrogenases, aldo/keto reductases have diverse roles in metabolism, one of which is neutralizing various toxic compounds [91,92]. Two more genes for formaldehyde detoxification were also upregulated in the Des_25 and Des_0 samples: a second glutathione-dependent formaldehyde dehydrogenase and a glutathione-independent formaldehyde dehydrogenase (Figure 6E,F). Unique to the Des_0 samples were the upregulation of an additional organic hydroperoxide resistance protein and superoxide dismutase (Figure 6F).

## 4. Discussion

The increasing intensity of biological stressors with altitude in the atmosphere [93,94] constrains the upper boundary of habitability in the Earth’s biosphere. For microbes that endure prolonged exposure to conditions in the stratosphere, UVR and desiccation are the most important parameters limiting survival [14,20,21,95,96,97,98,99,100,101,102,103,104,105,106]. *C. aetherium* L6-1 was isolated from aerosols collected at altitudes between 18 and 29 km ASL in New Mexico, USA, has previously been investigated for its high tolerance to UVCR and desiccation [14], and has recently been confirmed to be a phytopathogen of bean plants [45]. Attempts to establish a genetic system in *C. aetherium* using common methods of transformation and chromosomal modification were unsuccessful (Ellington, unpub. data). Therefore, we pursued several alternative approaches to examine the genetic and physiological basis of the high UVCR and desiccation tolerance in this bacterium.

Testing closely related species showed that none had a UVCR tolerance comparable to *C. aetherium* (Figure 1) and provided an opportunity for comparative genomics and directed evolution experiments. In addition, transcriptional profiling using RNA-seq provided insight into *C. aetherium*’s UV- and desiccation-stress responses. Given that both stressors produce DNA damage and generate ROS, we hypothesized that the high tolerances of *C. aetherium* L6-1 are due to having DNA repair and antioxidant genes not possessed by its more sensitive relatives.

### 4.1. Genetic Basis of UVR Tolerance in Curtobacterium

Previous studies have documented high UVR tolerance in members of the *Curtobacterium*. Sundin and Jacobs [107] examined bacterial UVCR tolerance in the phyllosphere of field-grown peanut plants and found that strains of *Curtobacterium* were the largest proportion (~43%, *n* = 213) of those with a UVCR minimum inhibitory dose (MID_c_) greater than 150 J m^−2^. Highly UVCR-resistant strains of *Curtobacterium* have also been isolated from desert rock varnish, with 17% remaining viable after a UVCR dose of 220 J m^−2^ [108]. While our results also suggest that UVCR tolerance is a common phenotype in this genus (i.e., the most UVCR “sensitive” species DSM 20129 has an LD_90_ of ~100 J m^−2^), they do not support the simplistic notion that the increased capacity to detoxify ROS alone is sufficient for a high UVCR tolerance (Figure 1B and Figure 3). The high resistance of DSM 20129 to oxidative stress is a curious phenotype, but given that ROS production by plants (i.e., during stress and in the regulation of immunity) and during desiccation is a common phenomenon, this trait may be a synapomorphy of the genus *Curtobacterium*. Notably, *C. aetherium* has a UVCR LD_90_ of 470 J m^−2^, which is more than twice the survival rate for other members of this genus and near that reported for *D. radiodurans* R1 (660 J m^−2^; [109]). In general, a species’ tolerance to UVR approximates natural exposure [24], which makes high tolerance to UVCR an intriguing phenotype for populations to maintain in surface environments. In any case, this phenotype is highly relevant for enabling survival and transport at high altitudes, where natural exposure to wavelengths of UVCR can occur [14].

Comparative genomic analyses with *C. flaccumfaciens* pv. *flaccumfaciens* DSM 20129 (ANI of 85% with L6-1 [45]) showed that *C. aetherium* possesses seven additional genes related to DNA repair and antioxidant systems (Figure 2; Appendix A): *nei2*, *uvrD2*, *pcrA*, *phr1*, *katA*, *perR*, and *nrdH2*. Two of these genes (*nei2* and *phr1*) were also found to be upregulated in response to UVR exposure. The *nei2* gene encodes a homolog of endonuclease VIII, a bifunctional DNA N-glycosylase and abasic (AP) site lyase involved in the BER pathway. This enzyme initiates DNA repair by recognizing and removing oxidative base lesions and cleaving the phosphodiester backbone of the resulting AP site [110,111]. The second gene, *phr1*, was annotated as a cryptochrome/photolyase family protein. The directed evolution experiments implicated a photolyase family protein in providing increased UVCR tolerance in strain DSM 20129 (Appendix A), prompting us to explore its photolyase further. While the mutated photolyase in the most tolerant evolved strain did not appreciably increase survival from the wild type when expressed in *E. coli*, this experiment did demonstrate that *C. aetherium* possesses a highly efficient photoreactivation system (Figure 4A).

In bacteria, UVR-induced DNA dimers are repaired through direct reversal by photolyases, or the lesion is removed via the NER pathway. NER requires the coordinated action of multiple proteins for the incision, unwinding, and excision of a damage-containing segment of DNA, followed by the resynthesis and ligation of the excised nucleotides [22,29,112]. Because NER is an energy-intensive process, the direct reversal of DNA photoproducts by photolyases may provide a more measured response to DNA damage, particularly when a generous supply of energy is not available to the cell. In a process termed photoreactivation, photolyases catalyze the direct reversal of pyrimidine dimers in DNA when activated with near-UV/blue light [78]. Two major classes of DNA photoproducts are produced by UVR: cyclobutane pyrimidine dimers (CPDs) and pyrimidine (6-4) pyrimidone photoproducts (6-4PPs). The direct repair of these lesions is catalyzed by photolyases specific to each lesion type [78,113].

Though CPD photolyases are widespread amongst bacteria, only three bacterial (6-4) photolyases have been characterized to date: PhrB of *Agrobacterium fabrum* [79,114] (formerly *Agrobacterium tumefaciens* [115]), CryB of *Cereibacter sphaeroides* [116,117] (formerly *Rhodobacter sphaeroides* [118]), and Vc(6–4) FeS-BCP of *Vibrio cholerae* [119]. Phylogenetic and structural comparisons with photolyases in the eukaryotic cryptochrome/photolyase family imply that prokaryotic (6-4) photolyases are more prevalent than initially thought and may represent the common ancestor of the cryptochrome/photolyase family [114]. Sequence alignment and predicted structural models of the *C. aetherium* photolyases (Appendix A) suggest that it possesses a putative CPD photolyase (*phr2*: KM842_RS16065) and (6-4) photolyase (*phr1*: KM842_RS01465). In contrast, *C. flaccumfaciens* pv. *flaccumfaciens* (DSM 20129) only possesses a CPD photolyase and is less efficient at photoreaction than *C. aetherium* (Figure 4A).

To demonstrate the functionality of *C. aetherium*’s putative 6-4PP and CPD photolyases, *phr1* and *phr2*, respectively, were heterologously expressed in *E. coli*, and their effects on its UVCR tolerance were examined. Though the expression of *phr1* alone did not appreciably increase UVCR tolerance over the control, the expression of *phr2* increased survival by ~325%, while the co-expression of *phr1* and *phr2* increased the survival rate to 400% (Figure 4B). These results are consistent with the larger number of CPD lesions (75–90% of total) generated upon UV exposure as compared to 6-4PP [22,120,121,122]. If further study demonstrates the *C. aetherium phr1* to encode a bona fide (6-4) photolyase, its activity alone might have a negligible effect on cell survival, since CPDs represent the overwhelming amount of DNA damage that occurs. However, when *phr1* and *phr2* are co-expressed, our data show that the capacity to repair both CPDs and 6-4PP dimers has a synergistic effect on survival. In addition to *phr1*, *phr2* was also upregulated upon exposure to UVCR (Figure 5C,D; Appendix A), indicating that these genes are part of the UV-resistome that assists *C. aetherium* with DNA damage repair. Oxidation of atmospheric organic compounds [123] and gene expression [124] in the aqueous phase of clouds have demonstrated that certain atmospheric conditions satisfy the requirements for microbial biochemical reactions to occur. Even so, most microbes aerosolized in the troposphere do not experience such favorable conditions, and those transported in the stratosphere are expected to be metabolically inert during transport [94]. However, if given an opportunity to function, the selective advantages of photoreactivation would make it a trait favorable to any species in an energy-limited, high-UVR-flux environment, including the atmosphere.

In addition to the canonical photolyases, the activation of a spore photoproduct lyase (SPL) family protein is a fascinating observation, given that no members of *Curtobacterium* form endospores and that this gene has not previously been described in non-spore-forming bacteria. SPL is a radical S-adenosyl-methionine (SAM) enzyme that repairs a specific UVR-induced dimer called spore photoproduct (5-thyminyl-5,6-dihydrothymine), and was initially believed to only be formed when bacterial endospore DNA is exposed to UVR [125,126]. The DNA within endospores is bound by small, acid-soluble spore proteins that alter its conformation from the typical B form to the A form [127]. While CPD and 6-4PP photolesions are the major dimers produced by UV irradiation in vegetative cells, the altered DNA conformation of endospores instead favors the production of spore photoproducts [128]. Interestingly, a similar B- to A-DNA conformational transition has been documented in non-spore-forming bacteria in response to desiccation [129], inducing the formation of the spore photoproduct when exposed to UVR [130]. *C. aetherium* does not possess the genetic capacity to form spores, but does exhibit exceptional desiccation tolerance [14]. The expression of SPL (*splB*) in *Bacillus subtilis* is not regulated by DNA damage but is instead part of the transcriptional program activated by sporulation-specific sigma factors [131]. In contrast, the expression of SPL in *C. aetherium* was only observed upon UVCR exposure, suggesting that its regulation in this bacterium may be DNA-damage-dependent. Clearly, our data indicate that the phylogenetic distribution of *splB* is not restricted to select members of the Bacillota, and exploring its role in the DNA repair of non-spore-forming, desiccation-tolerant bacteria is fertile territory for further study.

The largest increase in gene expression observed in response to UVCR was associated with error-prone DNA polymerases, with some being more than 50-fold upregulated (log_2_FC > 5.7) in the UVC + rec samples. This implies that translesion synthesis is a critical component of *C. aetherium*’s UVCR resistance. The activation of additional genes putatively annotated as potential DNA repair-involved proteins (e.g., PD-(D/E)XK nuclease family protein, GIY-YIG nuclease family protein, ATP-dependent DNA helicase, etc.) was also identified as a component of *C. aetherium*’s UV-resistome.

### 4.2. Desiccation Response in C. aetherium

While the transcriptional responses of bacteria to desiccation have not been extensively characterized, the mechanisms employed by cells during dehydration are known to be dependent on the duration and intensity of water loss [32,132]. Under desiccation stress, low water activity prevents biochemical reactions, and the cell enters a state of dormancy. Therefore, the physiological and molecular processes that enhance survival under prolonged desiccation must be enabled before cells become metabolically static. The genetic response of *C. aetherium* to desiccation stress is complex, evidenced by the number, magnitude, and functions of DEGs observed with decreasing water contents (Figure 6A). We hypothesize that the inverse relationship observed between the number of DEGs and water content indicates that *C. aetherium* has a measured response to water deficit. Consistent with previous observations in bacteria [32,132,133,134], ion and sugar transporters were among the first genes to be highly upregulated during dehydration. Genes for the biosynthesis of osmoprotectants, such as trehalose, and fatty acid catabolism were upregulated early on, coupled with a switch in energy production metabolism from glucose to fatty acids, allowing the diversion of glucose to trehalose synthesis [135,136]. Accordingly, the downregulation of genes for fatty acid biosynthesis and excessive energy-consuming functions, such as motility, was also observed.

DNA repair systems are another important contributor to desiccation resistance, and they were more strongly activated as water loss became more severe. Notably, the DNA repair systems activated during desiccation stress were distinct from those observed when exposed to UVCR. Whereas BER-, NER-, and HR-related genes were activated by UVCR exposure, these systems were not upregulated to the same degree in the desiccated samples. Instead, genes for the Ku protein and a non-homologous end-joining DNA ligase, known components of the bacterial NHEJ system [137], were the predominant upregulated DNA repair genes. The two extra non-LigA DNA ligases may also play a role in NHEJ, as was found for DNA ligase C1 in *M. smegmatis* [138]. Genetic studies of *M. smegmatis* have also shown that NHEJ-deficient mutants are sensitive to water deficit, suggesting that this pathway may represent a key mechanism for protecting DNA from double-strand breaks during desiccation [139].

### 4.3. Cross-Tolerance to Environmental Stressors

Desiccation and UV/ionizing radiation tolerance are frequently found to co-occur in extremophilic bacteria and archaea [140,141,142,143,144,145,146,147,148]. Given that the acute doses at which radio-tolerant species are resistant greatly exceed natural ionizing radiation sources on Earth, there is no evolutionary basis for ionizing radiation resistance to arise through natural selection. The demonstration of a genetic linkage of desiccation and ionizing radiation resistance in *D. radiodurans* R1 has provided a possible explanation for this paradox. The leading hypothesis for the prevalence of extreme ionizing radiation tolerance in desiccation-resistant bacteria is that efficient DNA repair mechanisms that evolved to compensate for damage from water loss are also highly effective at repairing DNA damage produced by exposure to ionizing radiation [149]. Likewise, there is no evolutionary pressure to maintain a high UVCR resistance in any modern surface environment on Earth. In contrast, water stress is a very common phenomenon in the biosphere, and the genetic and biochemical mechanisms that are known to enhance survival to desiccation (e.g., ROS detoxification) should also enhance tolerance to UVCR. It is also important to note that when *C. aetherium* is desiccated (25% RH) and exposed to UVCR, its LD_90_ increases to 1200 J m^−2^ [14], a value ~3-fold higher than that when water was not actively removed from cells (Appendix A). As such, the capacity to survive water deficit inherently protects *C. aetherium* against the indirect effects (i.e., ROS generation) of UVR exposure.

In addition to efficient pathways for DNA damage repair, ionizing and UVCR resistomes also possess mechanisms to physiologically detoxify the ROS generated during exposure to these high-energy wavelengths. ROS produced when cells are exposed to UVR or desiccated can oxidize lipids and proteins [150,151], as well as react with any other cellular constituent. ROS detoxification is known to have a role in bacterial desiccation resistance [151], and radio-sensitive strains of *D. radiodurans* are more susceptible to oxidative damage than radio-tolerant strains [75]. Given that oxidative stress is a major contributor to the damage caused by desiccation and radiation exposure, an efficient antioxidant system is expected to be a crucial trait of xero- and radio-tolerant species. Our experiments confirm this for *C. aetherium* and *C. flaccumfaciens* DSM 20129, with neither showing any significant increase in intracellular ROS concentration after UVCR exposure (Figure 3). Surprisingly, antioxidant-related genes were not found to be upregulated in *C. aetherium* after UVCR exposure, indicating that its antioxidant system is constantly poised to encounter oxidative stress. Other members of *Curtobacterium* may also possess this trait, which is a contention supported by a study that exposed phyllosphere communities to highly oxidative ozone (5000–10,000 ppb) and found that the relative abundance of *Curtobacterium* taxa increased ~4-fold, implying a high tolerance to ozone [152]. Consequently, it is tempting to speculate on the unique role that microbial antioxidant mechanisms could play during stratospheric transport, where the ozone concentration (~1000 to 8000 ppb at altitudes between 15 and 30 km ASL) can be ~1000-fold higher than in the air near the surface [153].

Under the experimental conditions we tested, antioxidant enzyme-encoding genes in *C. aetherium* were only upregulated in response to water loss, including glutathione peroxidase, hydroperoxide resistance protein, superoxide dismutase, and manganese catalase (Figure 6C–F). The upregulated glutathione-utilizing proteins are worth noting, as *C. aetherium* (and most Gram-positive bacteria) do not possess genes for glutathione synthesis. Instead, many actinobacteria produce a different thiol, mycothiol, which has been suggested to be a substitute for glutathione [43]. *C. aetherium* does possess genes for mycothiol synthesis (Appendix A), and thus it may function similarly as in other actinobacteria. In addition, high intracellular Mn^2+^ concentrations have been shown to contribute to desiccation and radiation resistance in microorganisms such as *D. radiodurans* [154,155]. *C. aetherium* upregulated metal ion transporters and downregulated a manganese efflux pump when desiccated, implying a potential role for Mn^2+^-dependent ROS scavenging under desiccation stress.

## 5. Conclusions

We investigated the genetic basis for extreme UVCR and desiccation resistance in a bacterium capable of surviving conditions in the stratosphere [14], where high UVCR fluxes and low water availability represent endmember values for these variables in the biosphere. Identifying the genetic underpinnings for the expressed characteristics mitigating damage from UVCR was facilitated by the rich history of experimental work that characterized the biochemistry and molecular biology of DNA repair processes in model bacterial species. The *Curtobacterium* genomes examined encode genes for most of the typical complement of DNA repair proteins documented in *E. coli* and *B. subtilis*. This implies that *C. aetherium* may be able to operate its canonical DNA repair pathways (e.g., photoreactivation and SPL) in a manner that is more effective than other species. Though mechanisms to cope with oxidative and water stress are invaluable to most microbes inhabiting terrestrial environments, the cross-protective effects of minimizing excessive ROS would also be effective for enduring the high UVR fluxes that intensify with altitude in the atmosphere.

## Figures and Tables

**Figure 1 microorganisms-13-00756-f001:**
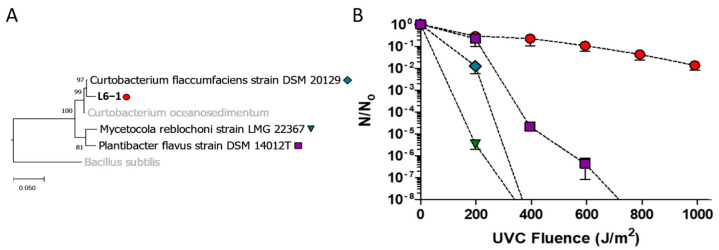
The phylogenetic relatedness among the bacterial strains and their tolerance to UVCR. (**A**) A maximum-likelihood analysis of full-length 16S rRNA gene sequences (aligned sequence = 1642 bp) from the stratospheric isolate *C. aetherium* L6-1 and the closely related strains used in this study. (**B**) UVCR tolerance, based on the surviving number of cells (N) at each UVCR dose divided by that in unexposed populations (N_0_). The plotted values are averages from three independent replicates, and the error bars represent SEM.

**Figure 2 microorganisms-13-00756-f002:**
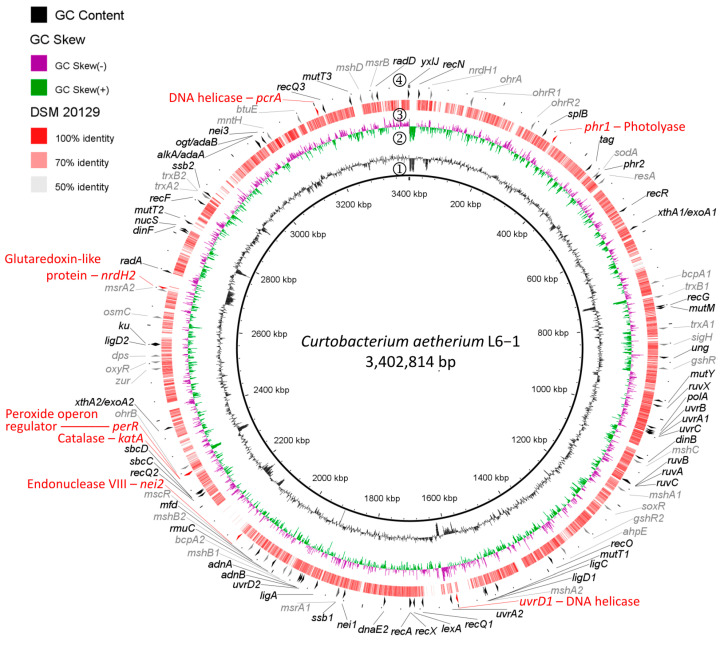
The DNA repair and ROS detoxification genes in the *Curtobacterium aetherium* L6-1 genome not found in strain DSM 20129.The genomes were compared using the BLAST Ring Image Generator (BRIG) with the blastp algorithm. The rings represent the following (from innermost to outermost): ① GC content, ② GC skew of the L6-1 genome, ③ percent identity to L6-1 of the protein orthologs found in strain DSM 20129, and ④ DNA repair (black) and ROS detoxification (gray) genes. The genes present in L6-1 but absent in DSM 20129 are indicated in red.

**Figure 3 microorganisms-13-00756-f003:**
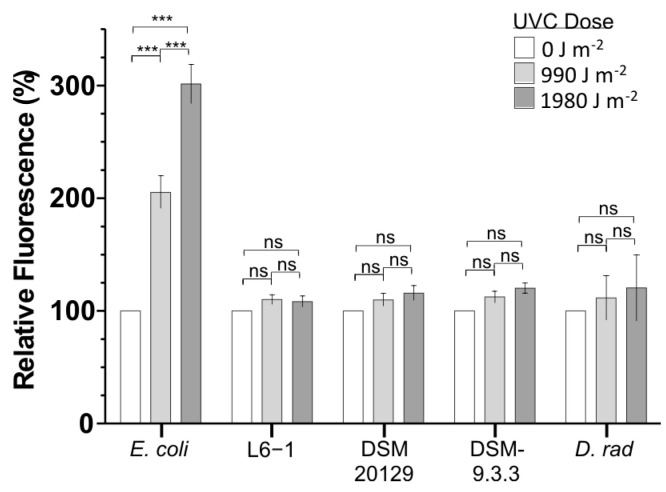
The cellular ROS concentrations in UVCR-treated cells. Intracellular ROS concentrations were quantified in the stratospheric isolate (*C. aetherium* L6-1), the DSM 20129 wild-type parent (*Curtobacterium flaccumfaciens* pv. *flaccumfaciens*), and UV-evolved strain (DSM-9.3.3) using the free radical probe H_2_DCFDA. The cells were exposed to UVCR for 5 and 10 min. to achieve the experimental dosage, and fluorescence intensity was compared with unexposed controls (t = 0, 100% fluorescence). *E. coli* MG1655 and *D. radiodurans* R1 were used as control strains. The data shown are the averages of three independent replicates, with the error bars indicating SEM. (ns = not significant, *** = *p*-value < 0.001; one-way ANOVA and Tukey’s post-test.)

**Figure 4 microorganisms-13-00756-f004:**
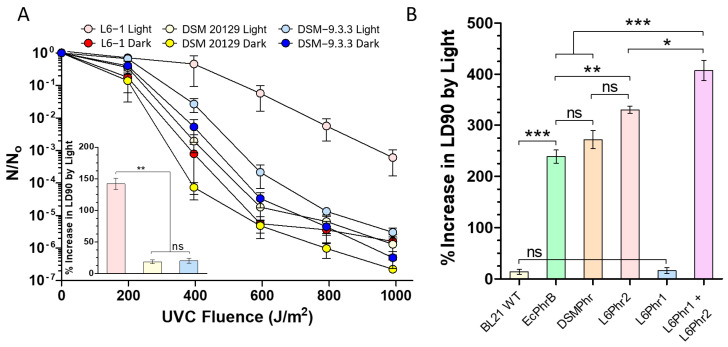
The photolyase activity in UVCR-treated cells. (**A**) The UVCR survival curve of the stratospheric isolate (*C. aetherium* L6-1), the DSM 20129 wild-type parent (*Curtobacterium flaccumfaciens* pv. *flaccumfaciens*), and the UV-evolved strain (DSM-9.3.3) under dark and photoreactivation (light) conditions. Inset: The percent increase in LD_90_ with 1 h of incubation in white light vs. dark after UVCR exposure. The average of four independent replicates is plotted, with the error bars representing SEM. (**B**) The percent increase in LD_90_ of *E. coli* BL21 cells expressing either the L6-1 *phr2* (L6Phr2), L6-1 *phr1* (L6Phr1), both L6-1 photolyase genes (L6Phr1 + L6Phr2), or the DSM 20129 *phr* (DSMPhr) with 1 h of incubation in white light versus dark after UVCR exposure. The expression of the *E. coli* CPD photolyase from wild-type BL21 cells (BL21 WT) and overexpression from the pDLx expression vector (EcPhrB) served as controls. The average of three independent replicates are plotted, with the error bars representing SEM. (ns = not significant; * = *p*-value < 0.05; ** = *p*-value < 0.01; *** = *p*-value < 0.001; one-way ANOVA and Tukey’s post-test.)

**Figure 5 microorganisms-13-00756-f005:**
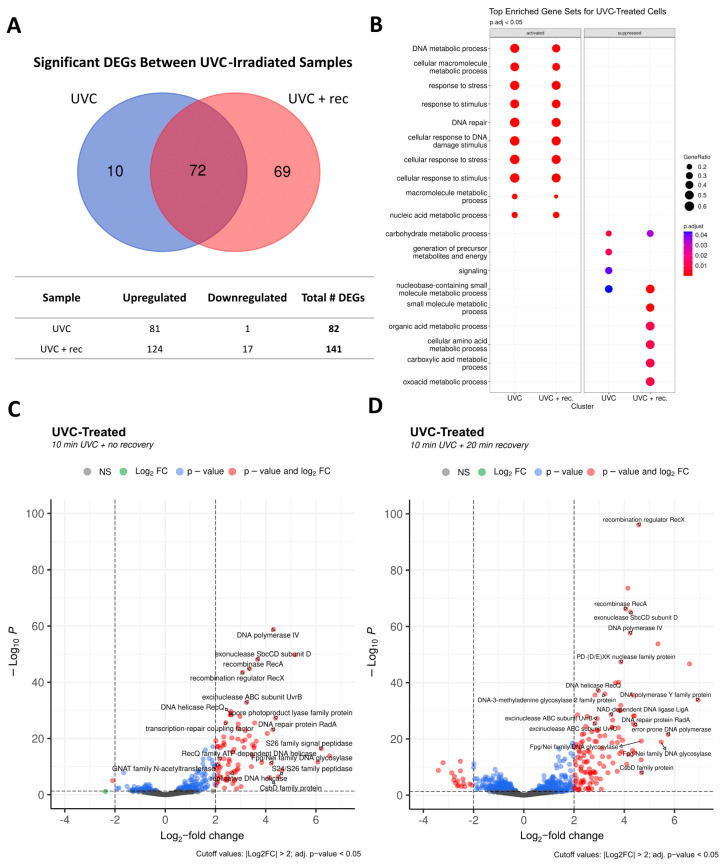
The differentially expressed genes in UVCR-treated cells. (**A**) A Venn diagram showing the number of DEGs unique to and shared between samples taken immediately after UVCR exposure and a 20 min recovery. (**B**) The top enriched gene sets for the UVC-treated cells. The dot size represents the ratio of genes activated (left) or suppressed (right) to the total within a category, with dot color representing *p*.adjust. Volcano plots displaying DEGs (**C**) immediately after UVCR exposure and (**D**) after a 20 min recovery period. Gene expression was determined to be significantly different if |log2FC| > 2 and *p*.adjust < 0.05. Points are colored gray for genes that were not differentially expressed, green if only the log2FC cutoff was met, blue if only the *p*.adjust cutoff was met, and red if both cutoff values were met.

**Figure 6 microorganisms-13-00756-f006:**
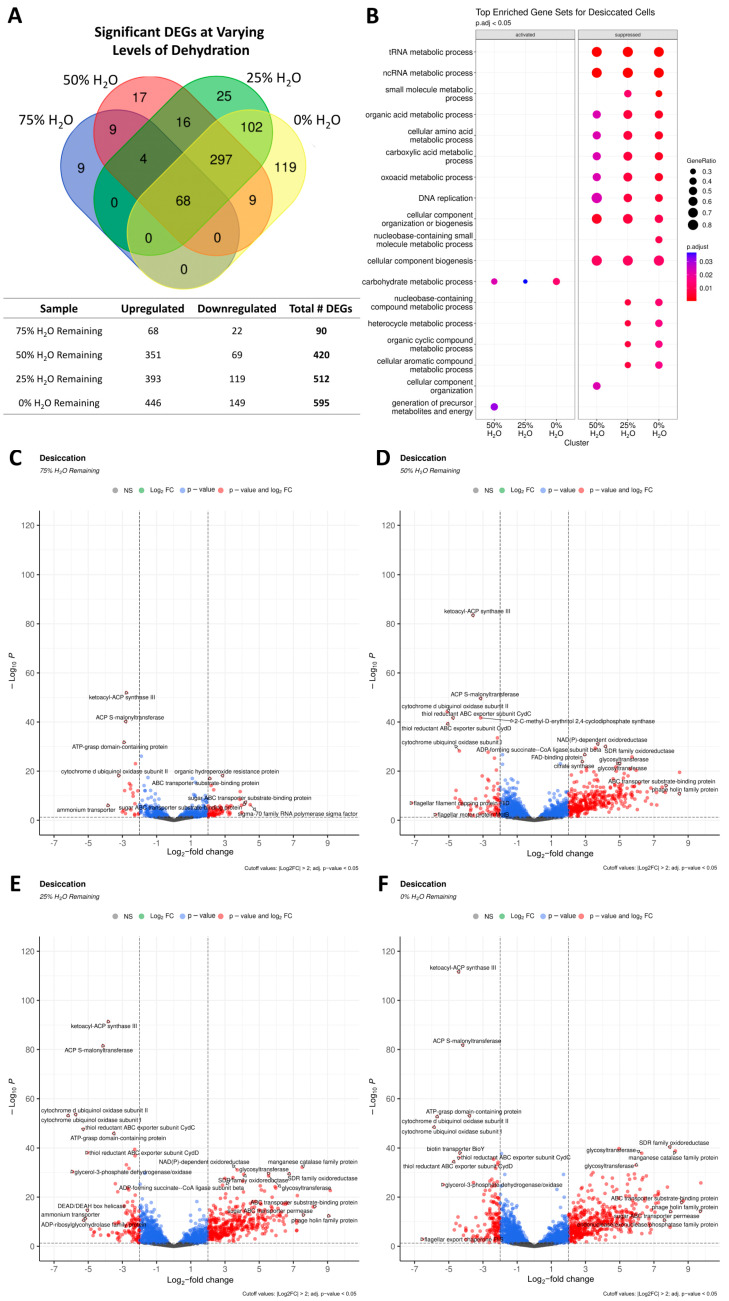
The differentially expressed genes in the cells under desiccation stress. (**A**) A Venn diagram showing the number of DEGs unique to and shared between samples taken at 75%, 50%, 25%, and 0% of the initial water content. (**B**) The top enriched gene sets for the desiccated cells. The dot size represents the ratio of genes activated (left) or suppressed (right) to the total within a category, with dot color representing *p*.adjust. Volcano plots displaying DEGs at (**C**) 75%, (**D**) 50%, (**E**) 25%, and (**F**) 0% of the initial water content. Genes were determined to be significantly differentially expressed if |log2FC| > 2 and *p*.adjust < 0.05. Points are colored gray for genes that were not differentially expressed, green if only the log2FC cutoff was met, blue if only the *p*.adjust cutoff was met, and red if both cutoff values were met.

**Table 1 microorganisms-13-00756-t001:** General genome features of the stratospheric isolate and its closest related reference strain.

Strain	Chr	Plasmids	Size (Mbp)	GC %	Genes	rRNA	tRNA	Proteins	ANI (±SD) %	AAI (±SD) %
*Curtobacterium aetherium* L6-1	1	0	3.4	72.0	3158	12	46	3072	83.64 ± 4.97	79.97 ± 13.57
*Curtobacterium flaccumfaciens* DSM 20129	1	1	3.8	70.9	3616	9	47	3517		

## Data Availability

The whole-genome sequences for *C. aetherium* L6-1 (CP076544.1; https://www.ncbi.nlm.nih.gov/nuccore/CP076544.1; 16 June 2021) and *C. flaccumfaciens* DSM 20129 (CP080395.1/CP080396.1; https://www.ncbi.nlm.nih.gov/nuccore/NZ_CP080395.1/; https://www.ncbi.nlm.nih.gov/nuccore/NZ_CP080396.1; 13 October 2024) are available in GenBank. The raw sequencing reads from the whole-transcriptome sequencing experiments are available in the NCBI Sequence Read Archive under BioProject PRJNA734343 (https://www.ncbi.nlm.nih.gov/bioproject/PRJNA734343/; 15 June 2021).

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
