# Peer review of "The Genetic Determinants of Extreme UV Radiation and Desiccation Tolerance in a Bacterium Recovered from the Stratosphere"

_microorganisms, 2025, doi:10.3390/microorganisms13040756_

Round 1

Reviewer 1 Report

Comments and Suggestions for Authors

This is a well designed and well written manuscript that describes several interesting DNA repair adaptations in this stratospheric bacterium.  I have no major editorial suggestions but a few questions for consideration:

  1. Are these bacteria really "surviving" in the atmosphere?  Are they biologically active under these conditions?  Is it fair to assume that what they do in LB at 30C is what they (could/might) do at 23 km?  
  2. Any speculation that the control and relatively UV-sensitive flaccumfaciens has lost the repair genes over time?  The paper seems to imply that L6-1 has gained extra genes but also mentions on p.18 that this group of organisms are generally all quite UVR.  It would be interesting to read some speculation on the ways in which selection has operated on L6-1.  Is it, in fact, the parent strain and 20129 (the "type" strain) the evolved, smaller genome strain in the absence of UVC-type environmental selection?
  3. The discrepancy between ROS and UV resistance in the control strain relative to the L6-1 or evolved strain (compare Figure 1 to Figure 3) is interesting and worthy of more discussion.  The argument being made in the paper seems to be that ROS resistance is correlated with UV resistance but that is clearly not the case for the DSM20129.   Why hasn't the "type" strain "lost" ROS resistance in the same manner it has "lost" UV resistance?  I presume the ROS selection in its natural environment is greater than the UV selection, certainly for UV-C.  Is the 20129 strain similarly exposed to dessication or the like?
  4. Is there any reason to think that Phr1 requires Phr2 to be active?  You suggest that Phr1 plays a minor role on 6-4s that is only detected in the presence of the more important Phr2 that is repairing the more numerous CPDs.  Would be interesting to measure rates of lesion removal using 6-4 and CPD specific mAbs.

Author Response

This is a well designed and well written manuscript that describes several interesting DNA repair adaptations in this stratospheric bacterium.  I have no major editorial suggestions but a few questions for consideration:

R1C1: Are these bacteria really "surviving" in the atmosphere?  Are they biologically active under these conditions?  Is it fair to assume that what they do in LB at 30C is what they (could/might) do at 23 km? 
Response: Many types of bacteria survive dispersal in the atmosphere, but given that cells lose water when aerosolized, it is generally assumed that metabolism is not maintained during airborne transport. A notable exception are the environmental conditions within clouds that have been shown to contain active microbial metabolisms (see discussion of this in lines 783-785). We have edited this portion of the discussion to clarify this important point.
The behavior of this bacterium under laboratory conditions provides good indications of its tolerance to and mechanisms mitigating the environmental stresses tested, but they do not mimic conditions at 23 km. For instance, when cells of C. aetherium are desiccated at low relative humidity, their resistance to UVCR is three times higher than fully hydrated cells (lines 873-877).

R1C2: Any speculation that the control and relatively UV-sensitive flaccumfaciens has lost the repair genes over time?  The paper seems to imply that L6-1 has gained extra genes but also mentions on p.18 that this group of organisms are generally all quite UVR.  It would be interesting to read some speculation on the ways in which selection has operated on L6-1.  Is it, in fact, the parent strain and 20129 (the "type" strain) the evolved, smaller genome strain in the absence of UVC-type environmental selection?
Response: Thank you for this comment. However, we are not able to speculate on the origin of these genes with confidence based on comparing the genomes of C. aetherium (L6-1) to C. flaccumfaciens (DSM 20129).  This would, however, be tenable if these species were members of a monophyletic and genomically coherent cluster. The genomes of these species have a relatively low ANI value (83%) and additional analyses (manuscript reference 45) indicate that they are affiliated with separate and distinct clusters within the genus Curtobacterium.  As such, it is challenging to establish ancestry in these groups with the genome data available and determine whether the absence of genes is due to them being lost or never acquired.

R1C3: The discrepancy between ROS and UV resistance in the control strain relative to the L6-1 or evolved strain (compare Figure 1 to Figure 3) is interesting and worthy of more discussion.  The argument being made in the paper seems to be that ROS resistance is correlated with UV resistance but that is clearly not the case for the DSM20129.   Why hasn't the "type" strain "lost" ROS resistance in the same manner it has "lost" UV resistance?  I presume the ROS selection in its natural environment is greater than the UV selection, certainly for UV-C.  Is the 20129 strain similarly exposed to dessication or the like?
Response: C. aetherium L6-1 is clearly more UVCR resistant than DSM 20129, but it is important to note that in comparison to many bacteria, DSM 20129 is more resistant to UVCR (lines 710-711). Even so, our results do not support the simplistic notion that increased capacity to detoxify ROS alone is sufficient for high UVCR tolerance. As far as hypothesizing on why DSM 20129 can deal efficiently with oxidative stress but has lost (or never gained) extreme UVCR resistance, there are a few possibilities worth considering. First, given that members of the Curtobacterium are widely distributed on plants (some, such as C. aetherium, are phytopathogens) and that ROS production in plants is linked to stress and regulation of plant immunity, this is a tempting connection to make. Secondly, water stress is a very common phenomenon in the types of terrestrial environments where Curtobacterium has been documented. In D. radiodurans, which is also extremely tolerant to UVC and ionizing radiation, its efficient ROS detoxifying mechanisms are thought to have evolved to combat desiccation stress (lines 864-869). We have added text to the revised manuscript that discusses these points (lines 710-716).

R1C4: Is there any reason to think that Phr1 requires Phr2 to be active?  You suggest that Phr1 plays a minor role on 6-4s that is only detected in the presence of the more important Phr2 that is repairing the more numerous CPDs.  Would be interesting to measure rates of lesion removal using 6-4 and CPD specific mAbs.
Response: We are not aware of research that has shown the activity of (6-4) photolyases (Phr1) to be dependent on the presence of CPD photolyases (Phr2) and agree that future work to confirm the specific types of DNA lesions they repair would be valuable. We did attempt to recombinantly express and purify C. aetherium’s photolyases in order to perform an in vitro photorepair assay. Unfortunately, we had difficulty purifying the proteins, and after numerous attempts to optimize purification conditions, we decided to proceed with the results of the in vivo assays for this manuscript. 

Reviewer 2 Report

Comments and Suggestions for Authors

The article presents comprehensive study of UV and desiccation resistance mechanisms of the bacterial strain isolated in stratosphere. Based on high novelty and significance of the work, as well as on the analysis of the results, reviewer may propose minor revisiosns:

  1. Introduction. Lines 100–115. In this paragraph, designate goal of the present study. In the present form, it is difficult to separate results of the work [45] and present article.
  2. 2.1. Bacterial Strains and Culture Conditions. C. aetherium L6-1 is the main subject of the study, Curtobacterium flaccumfaciens str. DSM 20129, Mycetocola reblochoni str. LMG 22367, and Plantibacter flavus str. DSM 14012T are closely related organisms. Could you explain the selection of other strains used in the sudy?
  3. Line 138. 5 “min. and” – Remove the point.
  4. Subsection 2.5 is entitled “2.5. Photolyase Activity Assays and Sequence/Structural Comparisons”, but does not contain the description of sequence and structure analysis.
  5. Table A3 and A4 are entitled “… genes found in Curtobacterium sp. L6-1 and Curtobacterium flaccumfaciens strain DSM 20129”. In the same time, the tables contain some genes, which are absent in both genomes. Therefore, Table titles may be changed.
  6. Section Results contains description of methods used in the study. These descriptions may be included in the section 2. Materials and Methods.

Author Response

The article presents comprehensive study of UV and desiccation resistance mechanisms of the bacterial strain isolated in stratosphere. Based on high novelty and significance of the work, as well as on the analysis of the results, reviewer may propose minor revisiosns:

R2C1: Introduction. Lines 100–115. In this paragraph, designate goal of the present study. In the present form, it is difficult to separate results of the work [45] and present article.
Response: Agreed. We edited this portion of the introduction to clearly state the objective of our study (lines 99-100).

R2C2: 2.1. Bacterial Strains and Culture Conditions. C. aetherium L6-1 is the main subject of the study, Curtobacterium flaccumfaciens str. DSM 20129, Mycetocola reblochoni str. LMG 22367, and Plantibacter flavus str. DSM 14012T are closely related organisms. Could you explain the selection of other strains used in the sudy?
Response: To enable a comparative approach, we identified the closest phylogenetic relatives to C. aetherium L6-1 that were available in public culture collections or from researchers we contacted with archived strains. We have edited this portion of the manuscript to clarify this (lines 121-125).

R2C3: Line 138. 5 “min. and” – Remove the point.
Response: Done

R2C4: Subsection 2.5 is entitled “2.5. Photolyase Activity Assays and Sequence/Structural Comparisons”, but does not contain the description of sequence and structure analysis.
Response: Thank you for pointing out this discrepancy. We edited the title of this section to accurately reflect its content.

R2C5: Table A3 and A4 are entitled “… genes found in Curtobacterium sp. L6-1 and Curtobacterium flaccumfaciens strain DSM 20129”. In the same time, the tables contain some genes, which are absent in both genomes. Therefore, Table titles may be changed.
Response: We have revised the titles of these tables as follows:
Table A3: DNA repair genes present and absent in Curtobacterium sp. L6-1 and Curtobacterium flaccumfaciens strain DSM 20129.
Table A4: ROS detoxification genes present and absent in Curtobacterium sp. L6-1 and Curtobacterium flaccumfaciens strain DSM 20129.

R2C6: Section Results contains description of methods used in the study. These descriptions may be included in the section 2. Materials and Methods.
Response: We agree that method descriptions do not belong in the results section, but are a bit confused by this comment because we are not certain what portion of the results the reviewer is referring to.

Reviewer 3 Report

Comments and Suggestions for Authors

Ellington and coworkers provide an excellent investigation into the UV and dessication resistance in an actinobacterium sample obtained from stratospheric aerosols. The research methods are very clearly described. The study provides quantitative, genetic, and bioinformatic data supported with robust statistical analyses to identify the genetic basis of dessication and UVR resistance. I am impressed with the depth of analysis conducted and presentation and discussion of results. The overall presentation of the study is excellent, and I believe it will be of great fit to the audience of Microorganisms.

I have a few comments for the authors to consider:

  1. In the introduction, I believe a separate paragraph discussing the study objectives would be helpful.
  2. Lines 350-351: I am curious why the ROS concentration in E.coli was so substantially different due to UVCR exposure.
  3. For Figures 5 and 6 - for the volcano plots, it would be good to have more distinctive colors. Currently, it is difficult to distinguish the grey from the green and blue, as all are lighter shades.
  4. Line 555: What is the reason behind the inverse relationship of DEG number with water availability?

Author Response

Ellington and coworkers provide an excellent investigation into the UV and dessication resistance in an actinobacterium sample obtained from stratospheric aerosols. The research methods are very clearly described. The study provides quantitative, genetic, and bioinformatic data supported with robust statistical analyses to identify the genetic basis of dessication and UVR resistance. I am impressed with the depth of analysis conducted and presentation and discussion of results. The overall presentation of the study is excellent, and I believe it will be of great fit to the audience of Microorganisms.

I have a few comments for the authors to consider:

R3C1: In the introduction, I believe a separate paragraph discussing the study objectives would be helpful.
Response: We appreciate this feedback and received a similar comment from Reviewer 2. The introduction has been revised to clearly indicate the objective of our study (lines 99-100).

R3C2: Lines 350-351: I am curious why the ROS concentration in E.coli was so substantially different due to UVCR exposure.
Response: E. coli does not possess highly efficient mechanisms to cope with oxidative stress, and as a result, ROS increases with UVR exposure. We inserted text that mentions this in lines 360-361. For D. radiodurans, its extraordinary antioxidant activity has been shown to be linked to low-molecular-weight Mn2+–metabolite complexes. Our study has provided initial findings on the specific mechanisms responsible for detoxifying ROS in C. aetherium, which can do so as efficiently as D. radiodurans under the conditions tested.  

R3C3: For Figures 5 and 6 - for the volcano plots, it would be good to have more distinctive colors. Currently, it is difficult to distinguish the grey from the green and blue, as all are lighter shades.
Response: Thank you for the thoughtful feedback regarding the color choices in our volcano plot. We selected the colors (red, blue, green and grey) because it is a scheme used widely to convey statistical significance and biological relevance in differential expression analyses. The color scheme is intuitive and aligns with standard practices in data visualization, facilitating quick and accurate interpretation by readers. Additionally, we ensured that the chosen colors are distinguishable for individuals with color vision deficiencies by selecting shades with sufficient contrast. Given these considerations, we believe that maintaining this color scheme enhances clarity and interpretability. However, if there are specific concerns regarding accessibility or readability, we would be happy to explore alternative palettes that preserve these advantages.

R3C4: Line 555: What is the reason behind the inverse relationship of DEG number with water availability?
Response: One hypothesis is that as water deficit increases, so too does cellular stress, which activates more genes associated with desiccation and to prepare for senescence. We have revised and added text that describes this interpretation of the results (lines 831-835).